# Early childhood caries intervention in Aboriginal Australian children: Follow-up at child age 9 years

Xiangqun Ju[1]*, Joanne Hedges[1], Dandara Gabriela Haag[1], Gustavo Hermes Soares[1], Lisa Gaye Smithers[2], Lisa M. Jamieson[1]

1 Australian Research Centre for Population Oral Health (ARCPOH), Adelaide Dental School, University of Adelaide, Adelaide, Australia, 2 School of Health and Society, University of Wollongong, Wollongong, Australia

* xiangqun.ju@adelaide.edu.au

## Abstract

### Objective

Dental caries is one of the most common preventable diseases among Indigenous children. The study aimed to estimate the efficacy of an Early Childhood Caries (ECC) intervention among Aboriginal Australian children over 9 years, and to explore potential risk factors associated with dental caries among Indigenous Australian children.

### Methods

Data were from a randomized controlled trial conducted in South Australia, Australia. Four hundred and forty-eight women pregnant with an Aboriginal child were randomly allocated to either an immediate (II) or delayed (DI) intervention group between January 2011 and May 2012. The immediate intervention comprised (1) provision of dental care to mothers during pregnancy; (2) application of fluoride varnish to teeth of children at ages 6, 12; and 18 months; (3) motivational interviewing delivered in conjunction with; and (4) anticipatory guidance. Mothers/children in the DI group received the same intervention commencing at child aged 2 years. Follow-ups occurred when children were aged 2-, 3-, 5-, 7- and 9-years. In this analysis, outcomes were severity of caries experience: mean dmft/DMFT at child aged 9 years. Dependent variables included mothers' baseline and seven years follow-up characteristics (age, education level, source of income, residential location, smoking and alcohol status) and children's birth and feeding, and dental behaviours characteristics (sex, gestation, birth weight, breastfeeding status and sweet food consumption, and frequency of tooth brushing). Multivariable log-Poisson regression models with robust standard error estimation were applied as a statistical model to estimate multivariable relationships

**Data availability statement:** The data cannot be shared publicly due to the participants' privacy concerns. Data are available from the University of Adelaide Data Access (contact via Australian Research Centre for Population Oral Health: arcpoh@adelaide.edu.au) for researchers who meet the criteria for access to confidential data.

**Funding:** This study was funded by the National Health and Medical Research Council of Australia (NHMRC, Project Grant 627350).

**Competing interests:** The authors have declared that no competing interests exist.

of dental caries and other covariates. Risk ratios (RRs) with their 95%CI were calculated. Sensitivity analyses were conducted by using the inverse-probability-of-censoring weighting (IPCW) to overcome the loss-follow-up issues.

## Results

Data were available for 367 (II = 180 and DI = 187) children at age 9 years. The mean dmft was 3.41 (95% CI: 2.95–3.87) and DMFT was 0.31(95%CI: 0.22–0.41). In multi-variable modelling, mean dmft was higher (RR = 1.13, 95% CI: 1.01–1.26) among DI children than II children, but there were no significant differences in the permanent dentition. Risk factors for caries severity in both the primary and permanent dentition included lower mothers' education level (<12 years level: dmft; RR = 1.56, 95% CI:1.31–1.86; and 'Trade or TAFT: DMFT: RR =3.40, 95%CI: 1.16-9.98). Other risk factors for dental caries experience in the primary dentition included preterm birth, low birth weight, child not breastfed and sugar consumption more than 10%, and in permanent dentition was self-rated 'fair/poor' or 'Good' children's oral health, compared with self-rated 'Excellent/very good' oral health.

## Conclusion

The present study suggests that, within this cohort, initiating an early childhood caries intervention during pregnancy and infancy may be associated with lower caries experience in the primary dentition by age 9 years compared to a later start. Low maternal education level was associated with caries severity in both primary and permanent dentitions. Sugar consumption, a modifiable risk factor, greater than 10% was an important contributor to dental caries in primary teeth.

## Introduction

Dental caries is one of the most common diseases to impact children at a global level; ranked 10th and 1st for the primary and permanent dentition, respectively [1,2]. The negative impact of dental caries on oral health-related quality of life of children and their parents has been observed [3,4]. These include child experiences of pain, difficulties in chewing and swallowing, difficulties concentrating in school settings and shame, and loss of workdays and/or financial stress relating to accessing dental care for parents. In Australia, national child oral health surveys estimate the prevalence of caries experience to be over 40% among children aged 5–10 years with primary teeth and under 25% among children aged 6–14 years in the permanent dentition [5].

Indigenous Australians are those who identify as being of Aboriginal and/or Torres Strait Islander descent. Aboriginal and Torres Strait Islander children make up 5.9% (an estimated 278,000) of the total child population in Australia [6]. They represent more than one-third of the total Indigenous population (34%). In the 2012–14 National Child Oral Health Survey, Indigenous children had approximately three times higher teeth with untreated decay in children aged 5–10 years (3.5 vs 1.2) and 8–14

years (0.8 vs 0.3), and had nearly two times higher the mean number of decayed, missing or filled tooth surfaces (DMFs/dmfs) in primary (2.9 vs 6.3) and permanent dentitions (0.7 vs 1.3) than their non-Indigenous counterparts [5,7].

Dental caries is a preventable chronic disease with multiple potential contributing factors. Maternal characteristics (including residing in regional and remote areas [8], low education and income, maternal smoking during pregnancy [9,10] and children's birth, feeding and oral health behavior characteristics (including low birthweight and preterm birth, consumption of free sugar food and beverages and brushing teeth less than twice daily [10,11]), have been positively associated with early childhood caries (ECC). ECC is the strongest predictor of dental caries in adulthood [12].

Motivational interviewing (MI) is an effective method [13] to modify knowledge and behaviours, which can be used as a brief intervention to increase motivation to improve people's oral health-related knowledge and behaviours, such as developing a good daily oral hygiene routine, reducing sugar consumption, smoking cessation, and alcohol advice, ultimately reducing the incidence of ECC [14]. However, Batliner and colleagues' study [15] found that MI intervention appeared to improve maternal knowledge but had no effect on oral health behaviors. Previous reports from this long-term study showed that a MI-based intervention was effective in preventing dental caries in the primary dentition of Aboriginal children [16–18]. However, the benefits of the intervention on the permanent dentition have not been tested. This aim of this study was to present data, collected at ages 7 and 9, from the participants of the "Baby Teeth Talk" study; estimate trends over time; test whether the early oral health benefits were sustained over time; and explore potential risk factors associated with dental caries among Indigenous Australian children.

## Methods

### Study design and sample size

The long-term 'Baby Teeth Talk (BTT)' study was an outcome assessor-blinded, closed-cohort cross-in randomized controlled trial (Registration ID: ACTRN12611000111976, the Australian New Zealand Clinical Trials Registry (ANZCTR)) conducted in South Australia, Australia. The BTT study randomly allocated 448 women, pregnant with an Aboriginal child, to either an immediate (II) or delayed (DI) intervention group between January 2011 and May 2012. The detail of how the participants were randomised has been previously reported [18,19]. Study staff were trained in MI by a registered member of the Motivational Interviewing Network of Trainers. They attended a basic two-day MI training course, followed by an intense one-day follow-up course. Monthly one-day follow-up training was continued for six months, followed by bi-monthly one-day coaching, and occasional ad-hoc telephone coaching, for another year [16].

The immediate intervention comprised: (1) provision of dental care to mothers during pregnancy; (2) application of fluoride varnish to teeth of children at ages 6, 12; and 18 months; (3) Multiple sessions of motivational interviewing (fidelity was scrutinised and found to be acceptable) [16] were delivered in conjunction with; (4) anticipatory guidance: in our study, tailored oral health educational packages were compiled with dental-specific information relevant for mothers (focus on dental care provision, pregnancy gingivitis) and for children (focus on first solid foods, caring for infant teeth when they first erupt, tooth brushing and fluoride, avoiding sugar-containing foods and beverages, baby's first dental check-up, eruption of molar teeth). Mothers/children in the DI group received the same intervention at 24, 30 and 36 months. After 36 months, the children were re-examined and had fluoride varnish applied at 5-, 7-, and 9-year-olds (**Fig 1**).

Ethics approval was obtained from the University of Adelaide Human Research Ethics Committee (H-057–2010), the Aboriginal Health Council of South Australia (04-09-362), the Government of South Australia and the Human Research Ethics Committees of the three participating South Australian birthing hospitals. The study additionally used the Ethical Conduct in Aboriginal and Torres Strait Islander Health Research guidelines to obtain consent. All participants provided written informed consent at each phase of the study. Meanwhile, written informed consent was obtained from the study child's parent, legal guardian, or next of kin to participate in the study.

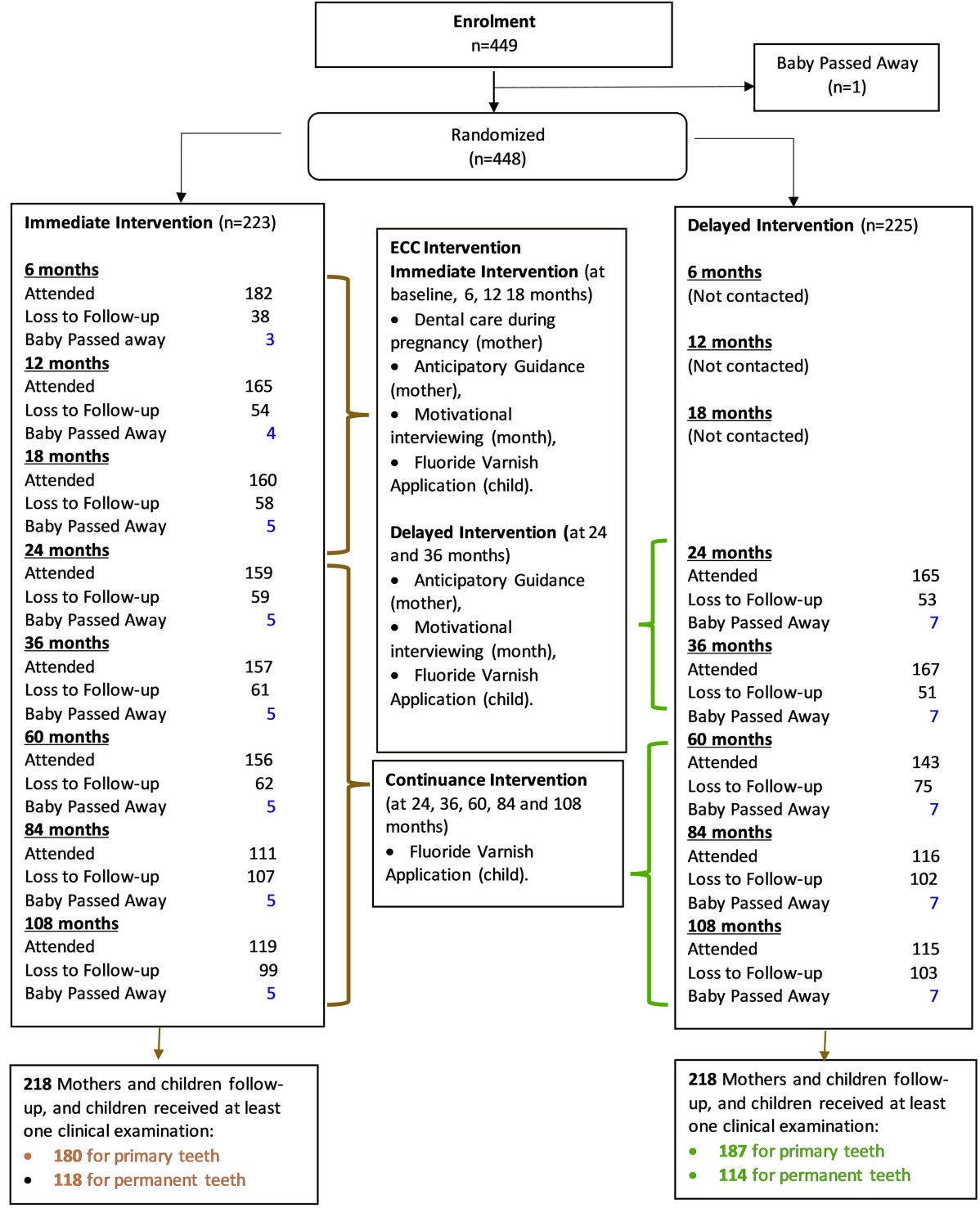

**Fig 1. Flow chart of participants through the key stages of the Randomised Clinical Trial.**

## Inclusivity in global research

Additional information regarding the ethical, cultural, and scientific considerations specific to inclusivity in global research is included in the Supporting Information (See Inclusivity in global research checklist).

## Data collection

Face-to-face interviews were conducted by experienced researchers, with three calibrated oral health professionals conducting oral examinations for children at ages 2, 3, 5, 7 and 9 years. Intra-class correlations for mean dt between each examiner and the gold standard examiner ranged from 0.80 to 0.88 [16].

## Variables

Outcome variables were severity of caries experience: the mean number of decayed (d/D), missing (m/M) and filled (f/F) teeth, and estimated dmft/ DMFT for primary and permanent dentitions, respectively. Because decayed and filled deciduous teeth were being exfoliated over the study period dmft was treated as an accumulated score – retaining the status of exfoliated teeth.

The exposure variable was early childhood caries (ECC) intervention status, which was identified as either immediate (II) or delayed (DI) group.

Covariates included:

1) mothers' baseline characteristics: maternal age, education level, source of income ('Job' or 'Centrelink': Centrelink is the Australian agency providing financial support for those who are unemployed), residential location, and smoking and alcohol status. Child variables included sex, gestational age, birth weight, breastfeeding, free sugar consumption of total energy intake at two years (This variable was calculated using dietary data collected at the 2-year follow-up, based on the average of one 24-hour dietary recall conducted by a trained researcher using a food model booklet, and, where available, up to two additional 24-hour diet diaries. Based on this, the variable was categorised as '<5%', '5%–10%', '11%–15%', or '>15%') [20], and frequency of tooth brushing.

2) mothers/primary carers and children's characteristics at seven years follow-up included primary carer's age, education level ('High school or less', 'Trade or TAFE: Technical and Further Education' or 'University'), employed status, source of income, and smoking and alcohol status (The two variables were identified from responses to 'What is your 'Smoking status' and 'What is your Alcohol drinking status'? These were then were catgorised as 1) Currently smoke/drink alcohol. 2) Used to smoke/drink alcohol, 3) Have never smoked/drunk alcohol). Study child's characteristics included self-rate general and oral health, sugar-sweetened beverage (SSB) consumption of total food intake (the proportion of SSB=(SSB scores/Dietary Guideline Index (DGI) scores)*100) [21] was derived from dietary data at 7-year follow-up, and was classified '<5%', '5%–10%', '11%–15%', or '>15%', and frequency of tooth brushing.

## Statistical analysis

The analysis began with the computation of univariate statistics describing the frequency and percentage of covariates and associated mean and 95% confidence intervals (CI) of dental caries. The intervention effects were also compared between the II and DI group. Multivariable log-Poisson regression models with robust standard error estimation were then applied as a statistical model to estimate multivariable relationships of dental caries and other covariates (including intervention status, mothers' and children's characteristics). Maternal and child characteristics were assessed at baseline and at the 2-year follow-up for the primary dentition analysis, and at the 7-year follow-up for the permanent dentition analysis. Risk ratios (RRs) with their 95%CI were calculated. The dependent variable of these models was dental caries experience (dmft/DMFT). Unadjusted models (Model 1): bivariate relationships of dental caries and other covariates. Mothers/primary

carers characteristics were entered in Model 2; Children's birth and feeding, dental behavior and free sugar or SSB consumption characteristics were added in Model 3, with the final model (Model 4) comprising all covariates. Sensitivity analyses were conducted by using the inverse-probability-of censoring weighting (IPCW) to overcome the loss-follow-up issues. Under the assumption that data were missing at random (MAR), IPCW was calculated and examined the balance of covariates. Data analyses were performed using SAS statistical software (SAS 9.4, SAS Institute Inc., Cary, NC, USA).

## Results

Of the 448 participants recruited at baseline, 12 babies who died before 2 years follow-up (5 in II and 7 in DI group), 84.2% children had at least one primary teeth and 53.2% received at least one permanent teeth examination at age 9 followed up. There was no difference in followup rates between the intervention groups. (Fig 1). Loss-to-follow-up was tracked, and reason included moving away from the study site, or becoming ill and unable to communicate.

### Dental caries in primary dentition

Mothers' baseline characteristics and children's birth and feeding characteristics at child aged 9 years follow-up is presented in Table 1. A higher proportion of pregnant mothers had an education level <12 years (more than 70%), received support from Centrelink (approximately 85%), were current smokers (50%), resided in non-metropolitan locations (approximately 65%), and used to drink alcohol (80%). A higher proportion of children were normal gestation (92%) and normal birth weight (90%), no breast feeding (58%), free sugar consumption 11% to 15% of total energy intake (56%) and less than twice/per day brushing teeth (71%).

The accumulated caries experience (dmft) trend in primary dentition from child aged 2–9 years is shown in **Fig 2**. The mean number of dmft increased from 0.77 to 3.41 (a more than 4.4 times increase). The mean number of dmft was higher in DI group (from 0.89 to 3.61) than in II group (from 0.64 to 3.20) at each follow-up time point (Fig 2a), but the growth rate of accumulated dmft was slowed in DI group (110%, 35.7%, 18.6% and 4.6%), compared to the II group (131%, 41.9%, 29.5%, 17.1%) from 2 to 9 years follow-up after receiving ECC intervention in both groups (Fig 2b). Also, the mean dt, mt, ft and dmft was shown in S1 Table.

The associations between dental caries experience (dmft) in the primary dentition and covariates at child aged 9 years are presented in Table 2. Higher mean dmft was observed among children in the DI group, whose mothers were in the younger maternal age group, education level < 12 years, Centrelink as source of income, non-metropolitan residential location, boys, free sugar consumption more than 5% and had low birth weight. After adjusting for all covariates, the mean dmft was 1.16 to 1.80 times higher among children whose mothers at baseline were in the younger maternal age group, had an education level less than 12 years, resided in non-metropolitan locations, used to consume alcohol, children who were preterm, had low birth weight, were not breast fed, had sweet food consumption more than 15% and between 11%−15% than their counterparts consuming less sweet foods/beverages. The mean dmft was 0.73 times lower among children whose mothers at baseline used to smoke tobacco than those who never smoked.

After the IPCW, similar results were observed in both unadjusted and adjusted models (Table 3), except the adjusted full model for people who brushed their teeth less than twice daily had a higher mean dmft than those who brushed their teeth more than twice daily. Also, the associations between untreated decay (dt) and filled teeth (ft) in the primary dentition and baseline covariates at child age 9 years are shown in S2 and S3 Tables.

### Dental caries in permanent dentition

Mothers/primary carers characteristics and children's characteristics at child aged 7 years follow-up is presented in Table 4. At seven years follow-up, a higher proportion of carers were aged less than 30 years (around 35%), 'Trade or TAFE' education level (nearly 45%), not employed (more than 60%) and received support from Centrelink (nearly 65%), and current smokers (more than 45%) and current drink alcohol (approximately 50%); and a higher proportion were children with

**Table 1. Baseline mother-child pairs characteristics by intervention status among Indigenous Australians at 9 years follow-up.**

| | All (n=367) | | II (n=180) | | DI (n=187) | |
|---|---|---|---|---|---|---|
| Baseline characteristics | N | % (95% CI) | N | % (95% CI) | N | % (95% CI) |
| **Mothers' characteristics** | | | | | | |
| **Maternal age** | | | | | | |
| 14-24 | 194 | 52.9 (47.7-58.0) | 101 | 56.1 (48.8-63.4) | 93 | 49.7 (42.5-57.0) |
| 25+ | 173 | 47.1 (42.0-52.3) | 79 | 43.9 (36.6-51.2) | 94 | 50.3 (43.0-57.5) |
| **Education** | | | | | | |
| ≤12 years | 258 | 70.7 (66.0-75.4) | 126 | 70.8 (64.0-77.5) | 132 | 70.6 (64.0-77.2) |
| >12 years | 107 | 29.3 (24.6-34.0) | 52 | 29.2 (22.5-36.0) | 55 | 29.4 (22.8-36.0) |
| **Source of Income** | | | | | | |
| Centrelink | 304 | 83.7 (79.9-87.6) | 147 | 82.6 (77.0-88.2) | 157 | 84.9 (79.7-90.1) |
| Job | 59 | 16.3 (12.4-20.1) | 31 | 17.4 (11.8-23.0) | 28 | 15.1 (9.9-20.3) |
| **Residential location** | | | | | | |
| Non-metropolitan | 229 | 63.1 (58.1-68.1) | 120 | 67.0 (60.1-74.0) | 109 | 59.2 (52.1-66.4) |
| Metropolitan | 134 | 36.9 (31.9-41.9) | 59 | 33.0 (26.0-39.9) | 75 | 40.8 (33.6-47.9) |
| **Smoking status** | | | | | | |
| Current | 175 | 50.4 (45.1-55.7) | 85 | 49.4 (41.9-57.0) | 90 | 51.4 (44.0-58.9) |
| Former | 86 | 24.8 (20.2-29.3) | 47 | 27.3 (20.6-34.1) | 39 | 22.3 (16.1-28.5) |
| Never | 86 | 24.8 (20.2-29.3) | 40 | 23.3 (16.9-29.6) | 46 | 26.3 (19.7-32.9) |
| **Alcohol status** | | | | | | |
| Current | 35 | 10.1 (6.9-13.3) | 19 | 11.0 (6.3-15.8) | 16 | 9.1 (4.8-13.5) |
| Used | 280 | 80.7 (76.5-84.9) | 137 | 79.7 (73.6-85.7) | 143 | 81.7 (75.9-87.5) |
| Never | 32 | 9.2 (6.2-12.3) | 16 | 9.3 (4.9-13.7) | 16 | 9.1 (4.8-13.5) |
| **Children's characteristics** | | | | | | |
| **Sex** | | | | | | |
| Male | 194 | 53.9 (48.7-59.1) | 95 | 54.0 (46.5-61.4) | 99 | 53.8 (46.5-61.1) |
| Female | 166 | 46.1 (40.9-51.3) | 81 | 46.0 (38.6-53.5) | 85 | 46.2 (38.9-53.5) |
| **Gestation** | | | | | | |
| Preterm | 24 | 7.6 (4.7-10.6) | 15 | 10.0 (5.1-14.9) | 9 | 5.5 (2.0-9.0) |
| Normal | 290 | 92.4 (89.4-95.3) | 135 | 90.0 (85.1-94.9) | 155 | 94.5 (91.0-98.0) |
| **Baby birth weight** | | | | | | |
| Low | 27 | 9.7 (6.2-13.3) | 15 | 11.4 (5.9-16.8) | 12 | 8.3 (3.7-12.8) |
| Normal | 250 | 90.3 (86.7-93.8) | 117 | 88.6 (83.2-94.1) | 133 | 91.7 (87.1-96.3) |
| **Breast feeding** | | | | | | |
| No | 197 | 57.9 (52.7-63.2) | 94 | 57.3 (49.7-65.0) | 103 | 58.5 (51.2-65.9) |
| Yes | 143 | 42.1 (36.8-47.3) | 70 | 42.7 (35.0-50.3) | 73 | 41.5 (34.1-48.8) |
| **Free sugar consumption of total energy intake** | | | | | | |
| > 15% | 35 | 13.5 (9.3-17.6) | 16 | 11.9 (6.4-17.5) | 19 | 15.1 (8.7-21.4) |
| 11%−15% | 145 | 55.8 (49.7-61.8) | 77 | 57.5 (49.0-65.9) | 68 | 54.0 (45.1-62.8) |
| 5%−10% | 62 | 23.8 (18.6-29.1) | 31 | 23.1 (15.9-30.4) | 31 | 24.6 (17.0-32.2) |
| < 5% | 18 | 6.9 (3.8-10.0) | 10 | 7.5 (3.0-12.0) | 8 | 6.3 (2.0-10.7) |
| **Tooth brushing** | | | | | | |
| < 2/day | 210 | 71.2 (66.0-76.4) | 103 | 70.5 (63.1-78.0) | 107 | 71.8 (64.5-79.1) |
| ≥ 2/day | 85 | 28.8 (23.6-34.0) | 43 | 29.5 (22.0-36.9) | 42 | 28.2 (20.9-35.5) |

Notes: II: Immediate intervention, DI: delayed intervention.

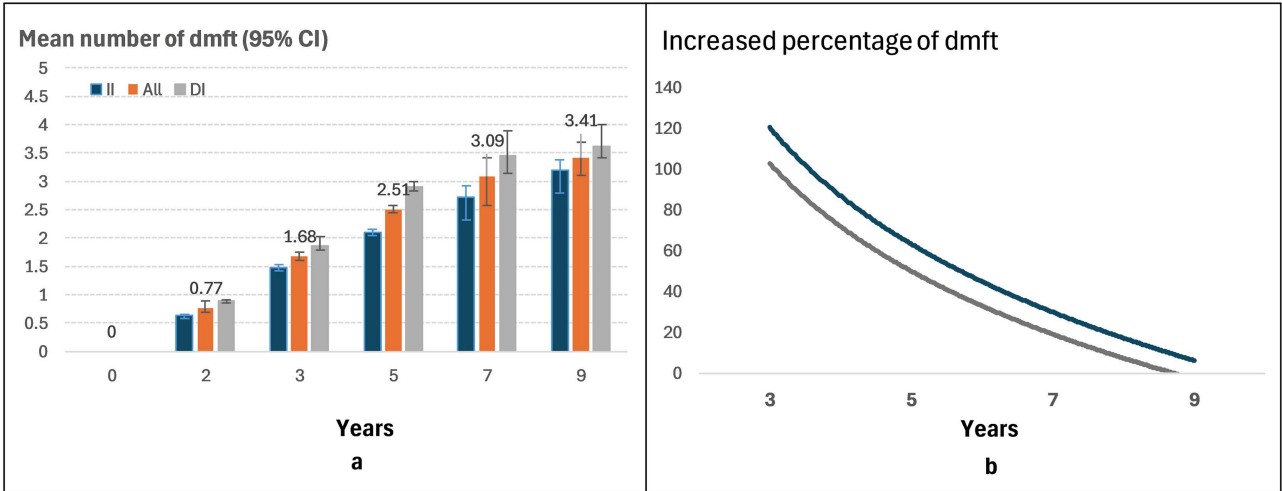

**Fig 2. Mean and increased percentage of caries experience (dmft) in primary dentition by Immediate intervention (II) and delayed intervention (DI) groups between child ages 2 and 9 years.**

self-rated excellent/very good general health (nearly 80%) and oral health (approximately 60%), SSB consumption 11% to 15% of total food intake (nearly 40%), and less than twice/per day brushing teeth (around 60%). A similar pattern was observed in both II and DI groups.

The accumulated caries experience (DMFT) trend in the permanent dentition from children aged 7–9 years is shown in **Fig 3**. The mean number of DMFT from the age of 7–9 years increased from 0.13 to 0.31 (a more than 2 times increase). The mean number of DMFT was higher (0.14) at 7 years but was lower (0.30) at 9 years in DI group, compared with II group (0.12 and 0.33, respectively) (Fig 3a), and the growth rate of accumulated DMFT was slowed in DI group (11.4%) than in II group (17.5%) from 7 years to 9 years follow-up (Fig 3b). Also, the mean DT, MT, FT and DMFT was shown in S1 Table.

The association between dental caries experience in permanent dentition (DMFT) and covariates in children aged 7–9 years is presented in Table 5. Higher mean DMFT was observed among children whose primary carers had 'Trade or TAFE' education, self-rated 'Good' general health and 'Fair/poor' and 'Good' oral health and brushing less than twice times daily; and lower mean DMFT were among children whose primary carer was not employed, received support from Centrelink and consumed SSB being 11% to 15% of total food intake. After adjusting for all covariates, the mean DMFT was more than 3 times higher among children whose primary carers had 'Trade or TAFE' education than those had 'University' education; 2–3 times higher among children who self-rated oral health were 'Fair/poor' and 'Good' than those with 'Excellent/very good' self-rated oral health; and nearly 2 times higher among children who brushed their teeth less than twice daily than those brushing twice or more times daily. After the IPCW, the similar results were observed in both unadjusted and adjusted models (Table 6), except crude model for people who brushed their teeth less than twice daily had higher the mean DMFT (RR = 1.93, 95% CI: 1.10–3.40) than those brushed their teeth more than twice daily. Also, the associations between dental caries experience (DMFT), untreated decay (DT) and filled teeth (FT) in the permanent dentition and baseline covariates at child aged 9 years are shown in S4–S6 Tables, and sensitivity analysis of association between DMFT and baseline mother-child pairs characterisers is presented in S7 table.

## Discussion

Our study showed that the ECC intervention delivered in early childhood had an effect on reducing dental caries experience in the primary dentition, but not the permanent dentition at child aged 9 years. Low maternal education level was

**Table 2. Models for the mean number of dmft at 9 years follow-up (RR, 95% CI).**

| | Model 1<br>RR (95% CI) | Model 2<br>RR (95% CI) | Model 3<br>RR (95% CI) | Model 4<br>RR (95% CI) |
|---|---|---|---|---|
| **Intervention group** | | | | |
| DI | *1.13 (1.01-1.26) | *1.19 (1.06-1.34) | 1.07 (0.93-1.23) | 1.11 (0.97-1.28) |
| II | ref | ref | ref | ref |
| **Mothers' characteristics at baseline** | | | | |
| **Maternal age** | | | | |
| 14-24 | *1.16 (1.03-1.30) | 1.02 (0.91-1.16) | | *1.16 (1.00-1.35) |
| 25+ | ref | ref | | ref |
| **Education level** | | | | |
| ≤12 years | ***1.62 (1.41-1.86) | **1.57 (1.35-1.82) | | **1.56 (1.31-1.86) |
| >12 years | ref | ref | | ref |
| **Source of Income** | | | | |
| Centrelink | **1.34 (1.13-1.59) | 1.07 (0.89-1.29) | | 0.97 (0.80-1.19) |
| Job | ref | ref | | ref |
| **Residential location** | | | | |
| Non-metropolitan | **1.61 (1.42-1.83) | **1.63 (1.43-1.86) | | ***1.84 (1.57-2.16) |
| Metropolitan | ref | ref | | ref |
| **Smoking status** | | | | |
| Current | 0.97 (0.85-1.11) | 0.95 (0.82-1.10) | | 0.82 (0.69-1.02) |
| Former | *0.71 (0.60-0.85) | *0.77 (0.65-0.92) | | *0.73 (0.60-0.89) |
| Never | ref | ref | | ref |
| **Alcohol status** | | | | |
| Current | 0.71 (0.55-1.08) | 1.05 (0.86-1.28) | | 1.18 (0.89-1.57) |
| Used | *0.72 (0.60-0.85) | 1.28 (0.99-1.66) | | 1.45 (1.02-2.05) |
| Never | ref | ref | | ref |
| **Children's characteristics** | | | | |
| **Sex** | | | | |
| Male | *1.12 (1.01-1.24) | | 1.06 (0.92-1.22) | 1.12 (0.97-1.30) |
| Female | ref | | ref | ref |
| **Gestation** | | | | |
| Preterm | 1.01 (0.80-1.28)) | | 1.11 (0.81-1.54) | *1.45 (1.02-2.05) |
| Normal | ref | | ref | ref |
| **Baby birth weight** | | | | |
| Low | **1.56 (1.19-2.05) | | *1.50 (1.13-2.09) | *1.35 (1.02-1.86) |
| Normal | ref | | ref | ref |
| **Breast feeding** | | | | |
| No | 1.02 (0.90-1.14) | | 0.99 (0.86-1.14) | *1.32 (1.01-1.78) |
| Yes | ref | | ref | ref |
| **Free sugar consumption of total energy intake** | | | | |
| > 15% | **1.53 (1.13-2.08) | | *1.52 (1.07-2.18) | **1.39 (1.15-3.71) |
| 11%−15% | **1.49 (1.17-1.89) | | *1.34 (1.02-1.76) | *1.25 (1.08-1.66) |
| 5%−10% | *1.28 (1.03-1.60) | | 1.16 (0.91-1.49) | 1.16 (0.90-1.50) |
| < 5% | ref | | ref | ref |
| **Tooth brushing** | | | | |
| < 2/day | 1.09 (0.95-1,25) | | *1.25 (1.04-1.51) | 1.16 (0.96-1.40) |
| ≥ 2/day | ref | | ref | ref |

Notes: RR: risk ratio, II: Immediate intervention, DI: delayed intervention; *P<0.05, **P<0.01, ***P<0.001.

Model 1: unadjusted model; Model 2: Adjusted for mothers/primary carers' characteristics; Model 3: adjusted for children's birth and feeding, dental behavior and free sugar consumption characteristics; Model 4: Full model, adjusting for all covariates.

**Table 3. Models using IPCW for the mean number of dmft at 9 years follow-up (RR, 95% CI).**

| | Model 1 | Model 2 | Model 3 | Model 4 |
|---|---|---|---|---|
| | RR (95% CI) | RR (95% CI) | RR (95% CI) | RR (95% CI) |
| **Intervention group** | | | | |
| DI | *1.13 (1.02-1.25) | *1.20 (1.08-1.33) | 1.04 (0.90-1.21) | 1.04 (0.90-1.21) |
| II | ref | ref | ref | ref |
| **Mothers' characteristics** | | | | |
| **Maternal age** | | | | |
| 14-24 | **1.17 (1.06-1.30) | 1.03 (0.92-1.15) | | 1.08 (0.92-1.26) |
| 25+ | ref | ref | | |
| **Education level** | | | | |
| ≤12 years | ***1.67 (1.46-1.90) | **1.60 (1.39-1.84) | | **1.45 (1.21-1.75) |
| >12 years | ref | ref | | ref |
| **Source of Income** | | | | |
| Centrelink | **1.37 (1.16-1.62) | 1.09 (0.91-1.31) | | 0.92 (0.75-1.12) |
| Job | ref | ref | | ref |
| **Residential location** | | | | |
| Non-metropolitan | ***1.59 (1.42-1.78) | **1.60 (1.42-1.80) | | ***1.62 (1.38-1.92) |
| Metropolitan | ref | ref | | ref |
| **Smoking status** | | | | |
| Current | 0.97 (0.86-1.10) | 0.95 (0.83-1.09) | | 0.98 (0.82-1.18) |
| Former | *0.73 (0.63-0.86) | *0.79 (0.67-0.93) | | *0.76 (0.62-0.95) |
| Never | ref | ref | | ref |
| **Alcohol status** | | | | |
| Current | 0.73 (0.58-1.02) | 0.80 (0.63-1.01) | | 0.43 (0.29-1.64) |
| Used | *0.73 (0.62-0.85) | *0.82 (0.69-0.97) | | *0.72 (0.56-0.91) |
| Never | ref | ref | | ref |
| **Children's characteristics** | | | | |
| **Sex** | | | | |
| Male | *1.11 (1.04-1.23) | | 0.92 (0.79-1.07) | 0.91 (0.78-1.06) |
| Female | ref | | ref | ref |
| **Gestation** | | | | |
| Preterm | 1.02 (0.83-1.27) | | 1.14 (0.82-1.58) | *1.26 (1.01-1.75) |
| Normal | ref | | ref | ref |
| **Baby birth weight** | | | | |
| Low | **1.60 (1.25-2.05) | | 1.28 (0.97-1.70) | 1.02 (0.75-1.37) |
| Normal | ref | | ref | ref |
| **Breast feeding** | | | | |
| No | 0.99 (0.89-1.11) | | 0.83 (0.72-1.02) | *1.19 (1.02-1.39) |
| Yes | ref | | ref | ref |
| **Free sugar consumption of total energy intake** | | | | |
| > 15% | **1.54 (1.17-2.04) | | **1.50 (1.09-2.08) | *1.27 (1.05-1.64) |
| 11%−15% | **1.51 (1.22-1.87) | | *1.37 (1.07-1.75) | *1.14 (1.09-1.43) |
| 5%−10% | *1.27 (1.04-1.55) | | 1.13 (0.90-1.41) | 1.12 (0.80-1.58) |
| < 5% | ref | | ref | ref |
| **Tooth brushing** | | | | |
| < 2/day | 1.10 (0.97-1.25) | | **1.34 (1.13-1.58) | *1.22 (1.02-1.45) |
| ≥ 2/day | ref | | ref | ref |

Notes: RR: risk ratio, IPCW: the inverse-probability-of-censoring weighting, II: Immediate intervention, DI: delayed intervention, *P<0.05, **P<0.01, ***P<0.001.

Model 1: unadjusted model; Model 2: Adjusted for mothers/primary carers' characteristics; Model 3: adjusted for children's birth and feeding, dental behaviour and free sugar consumption characteristics; Model 4: Full model, adjusted for all covariates.

**Table 4. Primary carer-child pairs characteristics at seven years follow-up by intervention status among Indigenous Australians.**

| | All (n = 232) | | II (n = 118) | | DI (n = 114) | |
|---|---|---|---|---|---|---|
| | N | % (95% CI) | N | % (95% CI) | N | % (95% CI) |
| **Primary carers characteristics at seven-year follow-up** | | | | | | |
| **Primary carers' age (years)** | | | | | | |
| < 30 | 78 | 33.6 (27.5-39.7) | 33 | 28.0 (19.7-36.2) | 45 | 39.5 (30.4-48.6) |
| 30-34 | 56 | 24.1 (18.6-29.7) | 33 | 28.0 (19.7-36.2) | 23 | 20.2 (12.7-27.7) |
| 35-39 | 48 | 20.7 (15.4-25.9) | 22 | 18.6 (11.5-25.8) | 26 | 22.8 (15.0-30.6) |
| ≥ 40 | 50 | 21.6 (16.2-26.9) | 30 | 25.4 (17.5-33.4) | 20 | 17.5 (10.5-24.6) |
| **Education level** | | | | | | |
| High school or less | 92 | 39.7 (33.3-46.0) | 43 | 36.4 (27.6-45.3) | 49 | 43.0 (30.4-48.6) |
| Trade or TAFE | 104 | 44.8 (38.4-51.3) | 58 | 49.2 (40.0-58.3) | 46 | 40.4 (31.2-49.5) |
| University | 36 | 15.5 (10.8-20.2) | 17 | 14.4 (8.0-20.8) | 19 | 16.7 (9.7-23.6) |
| **Employed status** | | | | | | |
| No | 141 | 61.0 (54.7-67.4) | 71 | 60.2 (51.2-69.1) | 70 | 61.9 (52.9-71.0) |
| Yes | 90 | 39.0 (32.6-45.3) | 47 | 39.8 (30.9-48.8) | 43 | 38.1 (29.0-47.1) |
| **Source of Income** | | | | | | |
| Centrelink | 147 | 63.4 (57.1-69.6) | 73 | 61.9 (53.0-70.8) | 74 | 64.9 (56.0-73.8) |
| Job | 85 | 36.6 (30.4-42.9) | 45 | 38.1 (29.2-47.0) | 40 | 35.1 (26.2-44.0) |
| **Smoking status** | | | | | | |
| Current | 107 | 46.3 (39.8-52.8) | 61 | 52.1 (43.0-61.3) | 46 | 40.4 (31.2-49.5) |
| Former | 63 | 27.3 (21.5-33.1) | 31 | 26.5 (18.4-34.6) | 32 | 28.1 (19.7-36.4) |
| Never | 61 | 26.4 (20.7-32.1) | 25 | 21.4 (13.8-28.9) | 36 | 31.6 (22.9-40.2) |
| **Alcohol status** | | | | | | |
| Current | 112 | 48.5 (42.0-55.0) | 64 | 54.7 (45.5-63.9) | 48 | 42.1 (32.9-51.3) |
| Used | 102 | 44.2 (37.7-50.6) | 46 | 39.3 (30.3-48.3) | 56 | 49.1 (39.8-58.4) |
| Never | 17 | 7.4 (4.0-10.8) | 7 | 6.0 (1.6-10.3) | 10 | 8.8 (3.5-14.0) |
| **Children's characteristics** | | | | | | |
| **Sex** | | | | | | |
| Male | 131 | 56.5 (50.0-62.9) | 66 | 55.9 (46.8-65.0) | 65 | 57.0 (47.8-66.2) |
| Female | 101 | 43.5 (37.1-50.0) | 52 | 44.1 (35.0-53.2) | 49 | 43.0 (33.8-52.2) |
| **Self-rated general health** | | | | | | |
| Fair/poor | 4 | 1.7 (0.0-3.4) | 3 | 2.6 (0.0-5.5) | 1 | 0.9 (0.0-2.6) |
| Good | 45 | 19.5 (14.3-24.6) | 20 | 17.1 (10.2-24.0) | 25 | 21.9 (14.2 −29.6) |
| Excellent/very good | 182 | 78.8 (73.5-84.1) | 94 | 80.3 (73.0-87.7) | 88 | 77.2 (69.4-85.0) |
| **Self-rated oral health** | | | | | | |
| Fair/poor | 27 | 11.7 (7.5-15.9) | 12 | 10.3 (4.7-15.8) | 15 | 13.2 (6.9-19.5) |
| Good | 68 | 29.4 (23.5-35.4) | 35 | 29.9 (21.5-38.3) | 33 | 28.9 (20.5-37.4) |
| Excellent/very good | 136 | 58.9 (52.5-65.3) | 70 | 59.8 (50.8-68.8) | 66 | 57.9 (48.7-67.1) |
| **% SSB consumption of total DGI at seven years follow-up** | | | | | | |
| > 15% | 55 | 24.6 (18.9-30.2) | 28 | 24.1 (16.2-32.0) | 27 | 25.0 (16.7-33.3) |
| 11%−15% | 86 | 38.4 (32.0-44.8) | 44 | 37.9 (29.0-46.9) | 42 | 38.9 (29.5-48.2) |
| 5%−10% | 26 | 11.6 (7.4-15.8) | 14 | 12.1 (6.1-18.1) | 12 | 11.1 (5.1-17.1) |
| < 5% | 57 | 25.4 (19.7-31.2) | 30 | 25.9 (17.8-34.0) | 27 | 25.0 (16.7-33.3) |
| **Tooth brushing** | | | | | | |
| < 2/day | 127 | 58.3 (51.7-64.9) | 64 | 57.1 (47.8-66.5) | 63 | 59.4 (49.9-68.9) |
| ≥ 2/day | 91 | 41.7 (35.1-48.3) | 48 | 42.9 (33.5-52.2) | 43 | 40.6 (31.1-50.1) |

Notes: II: Immediate intervention, DI: delayed intervention. SSB: Sugar-sweetened beverage; DGI: Dietary Guideline Index

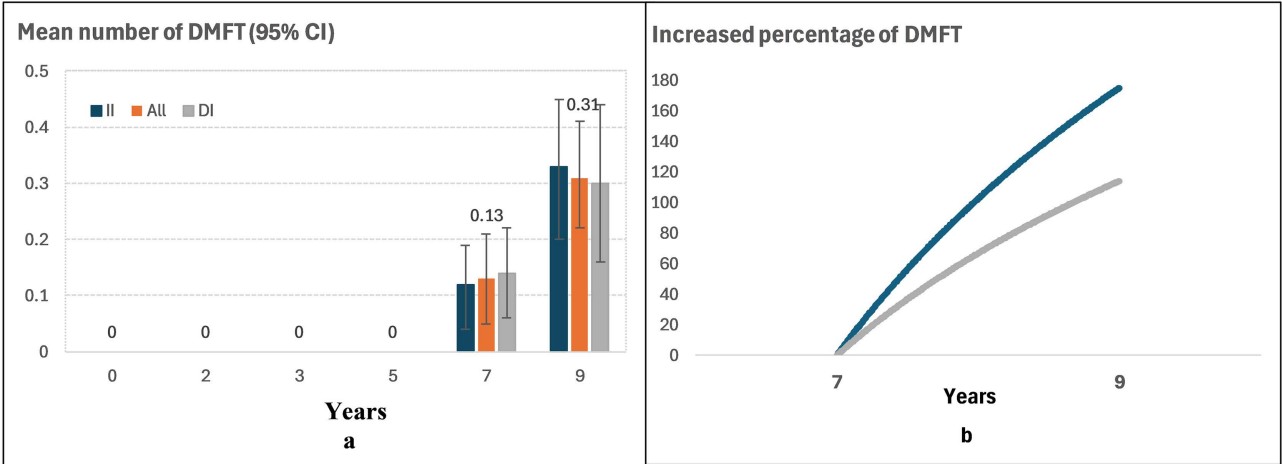

**Fig 3.  Mean and increased percentage of caries experience (DMFT) by Immediate intervention (II) and delayed intervention (DI) groups between children aged 7 and 9 years.**

associated with caries severity in both primary and permanent dentitions. Free sugar consumption greater than 10% was an important contributor to dental caries in primary teeth, a modifiable risk factor.

The findings revealed that the mean dmft at child aged 9 years (3.41) was higher among Indigenous South Australian children in our study than non-Indigenous Australian children (2.29), but lower than Indigenous Australian children at a national level (3.66) [22]. The mean DMFT (0.31) for children in our study at age 9 years was lower than both Indigenous (1.13) and non-Indigenous (0.61) Australian children aged 9 years [22], and a similar MI intervention effect was reported in another RCT study [23]. These findings indicate that the ECC intervention, which was community co-designed and culturally safe, may reduce dental caries in both the primary and permanent dentitions (although the permanent dentition was very similar across both II and DI group, even higher DMFT score was in II group (0.14) than in DI group (0.12)) in comparison to standard care. This is possibly because both II and DI groups received motivational interviewing at a very early stage, which provided not only increased motivation to improve parents' and children's oral hygiene behaviours, but also a framework for delivering diet, smoking cessation, and continuous use of fluoride varnish [24], results in reducing children's dental caries experience in both deciduous and permanent teeth. The high prevalence of missing teeth in the primary dentition [17], primarily due to early extractions from severe caries rather than natural exfoliation, contributed to the premature eruption of permanent teeth, thereby increasing the risk of dental caries in the permanent dentition. In addition, other clinical, sociodemographic and dental health-related behaviours factors, such as a higher prevalence of untreated caries in the primary teeth, lower family socioeconomic status, higher free sugar consumption, poor dental hygiene and irregular dental visits, are associated with caries incidence in the permanent dentition [25]. This result also reflects, from another perspective, that intervention for dental caries should be long-term and continuous to maintain its effectiveness.

This study is unique in global literature as this work is very hard to do by using culturally relevant ECC intervention involving MI among Indigenous Australians; and a randomised controlled trial design with substantial follow-up enabled not only the intervention effect to be estimated at child aged 9 years, but also the calculation of the time trend on dental caries of both primary and permanent dentitions; sensitivity analyses using IPCW to ensure unbiased estimation, although there were missing values in our study. Study limitations included not estimating time (age) when children were first introduced to free sugar and the number of cigarettes smoked by mothers during pregnancy [17].

**Table 5. Models for the mean number of DMFT at 9 years follow-up (RR, 95% CI).**

| | Model 1 | Model 2 | Model 3 | Model 4 |
|---|---|---|---|---|
| | RR (95% CI) | RR (95% CI) | RR (95% CI) | RR (95% CI) |
| **Intervention group** | | | | |
| DI | 0.78 (0.49-1.23) | 0.89 (0.54-1.43) | 0.78 (0.48-1.27) | 0.88 (0.52-1.49) |
| II | ref | ref | ref | ref |
| **Primary cares' characteristics at seven years follow-up** | | | | |
| **Primary carers' age (years)** | | | | |
| < 30 | 1.08 (0.61-1.93) | 1.15 (0.63-2.09) | | 1.28 (0.67-2.45) |
| 30-34 | 0.45 (0.20-1.01) | *0.40 (0.18-0.91) | | 0.47 (0.19-1.15) |
| 35-39 | 0.97 (0.49-1.90) | 0.85 (0.43-1.70) | | 1.16 (0.55-2.48) |
| ≥ 40 | ref | ref | | ref |
| **Education level** | | | | |
| High school or less | 2.46 (0.86-7.04) | **3.08 (1.04-9.17) | | 2.41 (0.77-7.59) |
| Trade or TAFE | ***3.67 (1.32-10.2) | ***3.71 (1.32-10.48) | | **3.40 (1.16-9.98) |
| University | ref | ref | | ref |
| **Employed status** | | | | |
| No | *0.47 (0.29-0.75) | *0.35 (0.17-0.74) | | 0.44 (0.19-1.05) |
| Yes | ref | ref | | ref |
| **Source of Income** | | | | |
| Centrelink | *0.58 (0.37-0.91) | 1.01 (0.51-2.00) | | 0.95 (0.43-2.09) |
| Job | ref | ref | | ref |
| **Smoking status** | | | | |
| Current | 1.70 (0.91-3.18) | *1.94 (1.00-3.77) | | 1.49 (0.75-2.98) |
| Former | 1.51 (0.76-3.01) | 1.60 (0.78-3.30) | | 1.36 (0.63-2.93) |
| Never | ref | ref | | ref |
| **Alcohol status** | | | | |
| Current | 1.91 (0.59-6.17) | 1.39 (0.42-4.63) | | 1.16 (0.33-4.04) |
| Used | 1.71 (0.52-5.57) | 1.68 (0.50-5.59) | | 1.36 (0.39-4.77) |
| Never | ref | ref | | ref |
| **Children's characteristics** | | | | |
| **Sex** | | | | |
| Male | 0.66 (0.42-1.05) | | *0.52 (0.30-0.88) | 0.64 (0.36-1.12) |
| Female | ref | | ref | ref |
| **Self-rated general health** | | | | |
| Fair/poor | 2.37 (0.58-9.75) | | 1.76 (0.38-8.14) | 0.77 (0.15-3.92) |
| Good | *1.70 (1.02-2.83) | | 1.15 (0.64-2.05) | 1.05 (0.57-1.95) |
| Excellent/very good | ref | | ref | ref |
| **Self-rated oral health** | | | | |
| Fair/poor | **3.00 (1.65-5.46) | | **3.82 (1.85-7.89) | **2.86 (1.35-6.04) |
| Good | **1.88 (1.11-3.17) | | *1.78 (1.00-3.18) | *1.86 (1.01-3.44) |
| Excellent/very good | ref | | ref | ref |
| **% SSB consumption of total DGI at seven years follow-up** | | | | |
| > 15% | 0.91 (0.51-1.63) | | 1.20 (0.62-2.31) | 1.03 (0.52-2.05) |
| 11%−15% | *0.45 (0.24-0.85) | | 0.83 (0.40-1.69) | 0.61 (0.28-1.30) |
| 5%−10% | 1.08 (0.53-2.15) | | 1.82 (0.84-3.96) | 1.28 (0.56-2.95) |

*(Continued)*

**Table 5.** (Continued)

| | Model 1 | Model 2 | Model 3 | Model 4 |
|---|---|---|---|---|
| | RR (95% CI) | RR (95% CI) | RR (95% CI) | RR (95% CI) |
| < 5% | ref | | ref | ref |
| **Tooth brushing** | | | | |
| < 2/day | **2.49 (1.43-4.34) | | *1.84 (1.01-3.33) | 1.77 (0.96-3.24) |
| ≥ 2/day | ref | | ref | ref |

Notes: RR: risk ratio, II: Immediate intervention, DI: delayed intervention; *P<0.05, **P<0.01, ***P<0.001. SSB: Sugar-sweetened beverage, DGI: Dietary Guideline Index.

Model 1: unadjusted model; Model 2: Adjusted for mothers/primary carers' characteristics; Model 3: adjusted for children's birth and feeding, dental behavior and SSB characteristics; Model 4: Full model, adjusted for all covariates.

## Maternal education level and children's dental caries

Maternal education is an important predictor of child dental caries, with findings from our study replicating previous investigations [26,27]. Low education is associated with low oral health literacy and knowledge, which in turn impacts understanding of the risk factors and behaviours related to dental caries, such as smoking, high consumption of sweet foods, irregular dental visiting and infrequent tooth brushing. Carer education level is positively associated with employment status and family income. Children from low- income families often have poor nutrition and foods that are high in free sugar or SBB [28], which predisposes children to development of dental caries. Level of education is highly correlated to residential location [29]. Higher educational attainment is less common among Indigenous adults residing in regional and remote areas [8] due to lack of quality education resources and limited access to education services (such as libraries and information technology).

## Sweet food intake and dental caries

Sugar consumption is one of the main causes of dental caries, both quantity (dose-response relationship) [2,30] and child ages when sugar consumption is initiated. Evidence shows that children with an early introduction to sugar (<12 months) had 1.5 times higher dental caries experience and untreated dental caries than children introduced to sugar at an older age (after 24 months) [31]. Sugar consumption before bedtime particularly increases the risk of caries [32]. This is due to reduced saliva flow and sustained low plaque pH later in the day, leading to erosion of tooth enamel and greater susceptibility to dental caries. Our findings support the World Health Organization (WHO) [33] recommendation of reducing the level of free sugar consumption to less than 5% −10% of total energy intake per day to prevent dental caries.

## Maternal smoking and children's caries

Evidence demonstrates that maternal smoking during pregnancy is significantly associated with dental caries in children [34,35] and is dose-related (children whose mothers smoked during pregnancy more than half pack/day were more likely to develop dental caries) [36]. One possible mechanism is that tobacco smoking affects the formation or mineralization of the primary teeth [37]. Maternal smoking during pregnancy is also harmful to the health of the foetus, leading to foetal malformations, shortened gestational age and low birth weight [38]. Preterm birth and low birth weight are positively associated with dental caries [39,40]. The structure of dental enamel can be affected in premature babies [41], which is likely to increase susceptibility to dental caries, while premature babies are often associated with low birth weight. Our study did not produce similar results to previous studies [34,35] that maternal smoking during pregnancy was significantly associated with dental caries in children.

**Table 6. Models using IPCW for the mean number of DMFT at 9 years follow-up (RR, 95% CI).**

| | Model 1 | Model 2 | Model 3 | Model 4 |
|---|---|---|---|---|
| | RR (95% CI) | RR (95% CI) | RR (95% CI) | RR (95% CI) |
| **Intervention group** | | | | |
| DI | 0.88 (0.59-1.31) | 0.85 (0.55-1.32) | 0.78 (0.50-1.21) | 0.87 (0.54-1.41) |
| II | ref | ref | ref | ref |
| **Primary cares' characteristics at seven years follow-up** | | | | |
| **Primary carers' age (years)** | | | | |
| < 30 | 1.21 (0.71-2.08) | 1.26 (0.73-2.18) | | 1.46 (0.80-2.67) |
| 30-34 | 0.50 (0.23-1.05) | *0.44 (0.21-0.94) | | 0.53 (0.23-1.20) |
| 35-39 | 1.10 (0.59-2.05) | 0.95 (0.51-1.80) | | 1.29 (0.64-2.59) |
| ≥ 40 | ref | ref | | ref |
| **Education level** | | | | |
| High school or less | 2.44 (0.91-6.50) | *3.07 (1.11-8.46) | | 2.44 (0.84-7.05) |
| Trade or TAFE | **3.72(1.43-9.67) | **3.75 (1.42-9.89) | | *3.45 (1.27-9.41) |
| University | ref | ref | | ref |
| **Employed status** | | | | |
| No | **0.45 (0.29-0.68) | *0.36 (0.19-0.68) | | 0.45 (0.21-1.02) |
| Yes | ref | ref | | ref |
| **Source of Income** | | | | |
| Centrelink | *0.55 (0.36-0.83) | 0.97 (0.52-1.80) | | 0.92 (0.45-1.89) |
| Job | ref | ref | | ref |
| **Smoking status** | | | | |
| Current | 1.63 (0.93-2.89) | *1.90 (1.04-3.46) | | 1.42 (0.75-2.67) |
| Former | 1.44 (0.76-2.71) | 1.56 (0.81-3.02) | | 1.39 (0.68-2.82) |
| Never | ref | ref | | ref |
| **Alcohol status** | | | | |
| Current | 1.94 (0.68-5.56) | 1.44 (0.49-4.23) | | 1.15 (0.37-3.53) |
| Used | 1.75 (0.61-5.05) | 1.74 (0.59-5.10) | | 1.32 (0.43-4.10) |
| Never | ref | ref | | ref |
| **Children's characteristics** | | | | |
| **Sex** | | | | |
| Male | *0.62 (0.41-0.92) | | *0.51 (0.32-0.83) | 0.64 (0.38-1.07) |
| Female | ref | | ref | ref |
| **Self-rated general health** | | | | |
| Fair/poor | 2.15 (0.60-7.75) | | 1.65 (0.41-6.50) | 0.72 (0.17-3.12) |
| Good | *1.62 (1.02-2.59) | | 1.10 (0.64-1.87) | 1.06 (0.60-1.88) |
| Excellent/very good | ref | | ref | ref |
| **Self-rated oral health** | | | | |
| Fair/poor | ***2.95 (1.71-5.09) | | **3.63 (1.88-7.02) | **2.64 (1.33-5.24) |
| Good | **1.86 (1.15-3.00) | | *1.71 (1.01-2.91) | *1.78 (1.01-3.13) |
| Excellent/very good | ref | | ref | ref |
| **% SSB consumption of total DGI at seven years follow-up** | | | | |
| > 15% | 0.49 (0.27-1.05) | | 1.15 (0.63-2.07) | 1.00 (0.54-1.86) |
| 11%−15% | 1.29 (0.68-2.45) | | 0.81 (0.42-1.55) | 0.58 (0.29-1.18) |
| 5%−10% | 1.12 (0.66-1.90) | | 2.02 (0.99-4.05) | 1.36 (0.64-2.89) |

*(Continued)*

**Table 6.** (Continued)

| | Model 1 | Model 2 | Model 3 | Model 4 |
|---|---|---|---|---|
| | RR (95% CI) | RR (95% CI) | RR (95% CI) | RR (95% CI) |
| < 5% | ref | | ref | ref |
| **Tooth brushing** | | | | |
| < 2/day | **2.63 (1.56-4.43) | | *2.00 (1.15-3.49) | *1.93 (1.10-3.40) |
| ≥ 2/day | ref | | ref | ref |

Notes: RR: risk ratio; IPCW: the inverse-probability-of-censoring weighting; II: Immediate intervention, DI: delayed intervention; *P<0.05, **P<0.01, ***P<0.001. SSB: Sugar-sweetened beverage, DGI: Dietary Guideline Index.

Model 1: unadjusted model; Model 2: Adjusted for mothers/primary carers' characteristics; Model 3: adjusted for children's birth and feeding, dental behaviour and SSB characteristics; Model 4: Full model, adjusted for all covariates.

## Conclusion

The present study suggests that, within this cohort, initiating an early childhood caries intervention during pregnancy and infancy may be associated with lower caries experience in the primary dentition by age 9 years compared to a later start. Mother's education level was negatively associated with dental caries in both primary and permanent teeth. Sugar consumption of more than 10% was an important contributor to dental caries in the primary dentition. The social and commercial determinants of health play an important role in shaping the oral health profile of Indigenous children through the life course.

## Supporting information

**S1 Table. Baseline mother-child pairs characteristics and dental caries among Indigenous Australians at 9 years follow-up.**
(DOCX)

**S2 Table. Models for the mean number of dt at 9 years follow-up (RR, 95% CI).**
(DOCX)

**S3 Table. Models for the mean number of ft at 9 years follow-up (RR, 95% CI).**
(DOCX)

**S4 Table. Models for the mean number of DMFT at 9 years follow-up (RR, 95% CI).**
(DOCX)

**S5 Table. Models for the mean number of DT at 9 years follow-up (RR, 95% CI).**
(DOCX)

**S6 Table. Models for the mean number of FT at 9 years follow-up (RR, 95% CI).**
(DOCX)

**S7 Table. Models using IPCW for the mean number of DMFT at 9 years follow-up (RR, 95% CI).**
(DOCX)

**S1 Checklist. CONSORT 2010 Checklist MS Word.**
(DOCX)

**S1 File. Inclusivity in global research questionnaire.**
(DOCX)

## Acknowledgments

The authors gratefully acknowledge the support of Baby Teeth Talk (Australia) study participants, study staff and partners: South Australian Dental Service, Colgate Palmolive, Women's and Children's Hospital, Lyell McEwen Hospital, Flinders' Medical Centre, Aboriginal Family Support Services, Aboriginal Primary Health Unit, Metro Aboriginal Family Birthing Program, Kura Yerlo Centre, Aboriginal Legal Rights Movement, Wodlitinattoai Program, Ninko Kutangga Patpangga, Kanggawodli Step Down Service, Kaurna Plains, Fleurieu Families, Gilles Plains Community Health, MADEC Community Support Services, Naporendi Aboriginal Forum, Nunga MiMinar, Aboriginal Strategy Unit of Families South Australia, Inbarendi College, Para West Adult Campus, Pangula Mannanurna Aboriginal Health Corporation and the Muno Parra Medical Centre, Second Story, Inner Southern Health Service, The Corner House, Louise Place, PARKS, Talking Realities Program, Child Youth Women's Health Services, Southern Junction Community Services, TAFE campuses, GP Plus centres, employment programs, childcare centres, women's centres, domestic violence shelters, and primary and high schools.

## Author contributions

**Conceptualization:** Xiangqun Ju, Lisa Gaye Smithers, Lisa M Jamieson.

**Data curation:** Joanne Hedges, Lisa Gaye Smithers, Lisa M Jamieson.

**Formal analysis:** Xiangqun Ju.

**Funding acquisition:** Lisa Gaye Smithers, Lisa M Jamieson.

**Investigation:** Joanne Hedges, Lisa Gaye Smithers, Lisa M Jamieson.

**Methodology:** Xiangqun Ju, Dandara Gabriela Haag, Gustavo Hermes Soares, Lisa Gaye Smithers, Lisa M Jamieson.

**Project administration:** Joanne Hedges, Lisa M Jamieson.

**Software:** Xiangqun Ju.

**Supervision:** Lisa Gaye Smithers, Lisa M Jamieson.

**Validation:** Joanne Hedges, Gustavo Hermes Soares, Lisa Gaye Smithers, Lisa M Jamieson.

**Visualization:** Dandara Gabriela Haag, Gustavo Hermes Soares, Lisa M Jamieson.

**Writing – original draft:** Xiangqun Ju.

**Writing – review & editing:** Xiangqun Ju, Joanne Hedges, Dandara Gabriela Haag, Gustavo Hermes Soares, Lisa Gaye Smithers, Lisa M Jamieson.

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
