## [Decision Letter · Decision Letter 0]

2 Apr 2025

PONE-D-24-53777Early Childhood Caries Intervention in Aboriginal Australian Children: follow-up at child age 9 yearsPLOS ONE

Dear Dr. Ju,

Thank you for submitting your manuscript to PLOS ONE. After careful consideration, we feel that it has merit but does not fully meet PLOS ONE’s publication criteria as it currently stands. Therefore, we invite you to submit a revised version of the manuscript that addresses the points raised during the review process.

**The authors are advised to give attention of methodological rigour. RCT provide the best form of evidence, however they require rigorous methods. The methods should be reproducible. Please pay attention to the reviewers and re-submit a point-by-point response in addition to revising the paper. Looking forward toa revised manuscript.**

We look forward to receiving your revised manuscript.

Kind regards,

Yolanda Malele-Kolisa, BDS, MPH, MDent, PhD

Academic Editor

PLOS ONE

Journal Requirements:

4. In the online submission form you indicate that your data is not available for proprietary reasons and have provided a contact point for accessing this data. Please note that your current contact point is a co-author on this manuscript. According to our Data Policy, the contact point must not be an author on the manuscript and must be an institutional contact, ideally not an individual. Please revise your data statement to a non-author institutional point of contact, such as a data access or ethics committee, and send this to us via return email. Please also include contact information for the third party organization, and please include the full citation of where the data can be found.                                                                                      

Reviewers' comments:

Reviewer's Responses to Questions

**Comments to the Author**

1. Is the manuscript technically sound, and do the data support the conclusions?

Reviewer #1: Partly

Reviewer #2: Yes

2. Has the statistical analysis been performed appropriately and rigorously? 

Reviewer #1: Yes

Reviewer #2: Yes

3. Have the authors made all data underlying the findings in their manuscript fully available?

Reviewer #1: No

Reviewer #2: Yes

4. Is the manuscript presented in an intelligible fashion and written in standard English?

Reviewer #1: No

Reviewer #2: Yes

5. Review Comments to the Author

Reviewer #1: The study is interesting, but several aspects require clarification.

1. How exactly was "sugar-sweetened beverage (SSB) consumption as a percentage of total food intake" or "percentage of free sugar consumption as a proportion of total energy intake" assessed? Which of these variables was included in the analysis?

2. On what basis was alcohol consumption categorized for women? What threshold was used to assign them an alcohol consumption status?

3. Why do the authors conclude that their study indicates that stopping smoking during pregnancy can reduce the incidence of ECC? The evidence for this statement is unclear.

4. The following sentence is too long and unclear:

"It cannot be denied that the prevalence of missing teeth due to dental caries in the primary dentition was much higher due to early extractions or exfoliation, resulting in premature eruption of permanent teeth, which increased the risk of dental caries in the permanent dentition."

Could the authors clarify this statement? Additionally, why do they suggest that the prevalence of missing teeth due to dental caries is higher due to exfoliation? Naturally exfoliated teeth are not included in the caries index, and severely carious teeth often do not exfoliate naturally.

5. Based on the results of the current study, it is not possible to conclude that "an ECC intervention delivered in early childhood had a long-term effect on reducing dental caries experience in the primary dentition at child age 9 years." There was no control group without intervention, and a comparison with national data alone is insufficient to confirm the intervention's success.

Reviewer #2: This paper is a very valuable report on the effectiveness of early intervention in preventing dental caries in indigenous Australian children up to the age of 9 years.

However, the following points require reconsideration.

The authors have already reported results up to 5 years of age in another journal (ref. 18), and more consideration should be given to the effect on caries in the permanent dentition.

Why is early permanent tooth caries more common in group II?

Why is there more caries in boys in the deciduous dentition but more in girls in the permanent dentition?

Need to explain Model in Tables 2 and 3.

6. PLOS authors have the option to publish the peer review history of their article (what does this mean?). If published, this will include your full peer review and any attached files.

Reviewer #1: No

Reviewer #2: No

---

## [Author Response · Author response to Decision Letter 1]

10 Jun 2025

Response to reviewers

Reviewer #1:

The study is interesting, but several aspects require clarification.

1. How exactly was "sugar-sweetened beverage (SSB) consumption as a percentage of total food intake" or "percentage of free sugar consumption as a proportion of total energy intake" assessed? Which of these variables was included in the analysis?

1) The percentage of total energy intake from free sugar was calculated using dietary data collected at the 2-year follow-up, based on the average of one 24-hour dietary recall conducted by a trained researcher using a food model booklet, and, where available, up to two additional 24-hour diet diaries. We have provided further details on how this was assessed (Line 167-171, page 9). This variable was included in the analysis of dmft for primary dentition (Line 191-193, page 10)

2) The proportion of sugar-sweetened beverage (SSB) consumption relative to total food intake was derived from dietary data collected at the 7-year follow-up (Line 179-182, page 9) and was included in the DMFT analysis for permanent dentition (Line 191- 192, page 10).

2. On what basis was alcohol consumption categorised for women? What threshold was used to assign them an alcohol consumption status?

Alcohol consumption was identified from responses to ‘What is your Alcohol drinking status?’ Then it was categorised as 1) Currently drink alcohol. 2) Used to drink alcohol, 3) Have never drunk alcohol. We have added this information in the Methods section (Line 175-178, Page 9).

3. Why do the authors conclude that their study indicates that stopping smoking during pregnancy can reduce the incidence of ECC? The evidence for this statement is unclear.

Many thanks. We have deleted this unclear sentence (Line 408, page 24).

4. The following sentence is too long and unclear:

"It cannot be denied that the prevalence of missing teeth due to dental caries in the primary dentition was much higher due to early extractions or exfoliation, resulting in premature eruption of permanent teeth, which increased the risk of dental caries in the permanent dentition."

Could the authors clarify this statement? Additionally, why do they suggest that the prevalence of missing teeth due to dental caries is higher due to exfoliation? Naturally exfoliated teeth are not included in the caries index, and severely carious teeth often do not exfoliate naturally.

Thanks. We have rewritten the sentence as ‘The high prevalence of missing teeth in the primary dentition—primarily due to early extractions from severe caries rather than natural exfoliation—contributed to the premature eruption of permanent teeth, thereby increasing the risk of dental caries in the permanent dentition (Line 354-357, Page 21-22).

5. Based on the results of the current study, it is not possible to conclude that "an ECC intervention delivered in early childhood had a long-term effect on reducing dental caries experience in the primary dentition at child age 9 years." There was no control group without intervention, and a comparison with national data alone is insufficient to confirm the intervention's success.

Thank you for your comments. Although we cannot definitively conclude that early childhood caries (ECC) intervention has a long-term effect on reducing dental caries experience, the data suggest a potential benefit. Specifically, the mean dmft was lower in the immediate intervention (II) group (3.20) compared to the delayed intervention (DI) group (3.61) at 9 years (S1_Table), indicating a possible positive impact of early intervention.

We have rewritten the sentence as ‘The present study suggests that an early childhood ECC intervention may have a positive impact on reducing dental caries experience in the primary dentition by age 9 years’ (Line 412-413, Page 24, and Abstract).

Reviewer #2:

This paper is a very valuable report on the effectiveness of early intervention in preventing dental caries in indigenous Australian children up to the age of 9 years.

However, the following points require reconsideration.

1. The authors have already reported results up to 5 years of age in another journal (ref. 18), and more consideration should be given to the effect on caries in the permanent dentition.

Thanks, we have added more consideration in the Discussion section (Line 361-362, Page 22).

2. Why is early permanent tooth caries more common in group II?

Early permanent tooth caries is more common in Group II, possibly because early intervention was no longer effective by the time the permanent teeth erupted. Additionally, the DI group received the same Early Childhood Caries (ECC) intervention after a 2-year follow-up, which may have provided more opportunities for preventive measures to influence the development of caries in permanent teeth.

3. Why is there more caries in boys in the deciduous dentition but more in girls in the permanent dentition?

This pattern may be explained by gene-by-sex interactions influencing caries experience in both primary and permanent dentitions (ref #1): ‘In the primary dentition, the magnitude of the effect of genes was greater in males than females. In the permanent dentition, different genes may play important roles in each of the sexes’.

Also, our findings align with those of a previous study (ref #2), which reported that males had a higher overall mean dmft and a greater number of untreated decayed primary teeth, while females exhibited higher DMFT scores in the permanent dentition (See S1_Table).

Reference

1. Shaffer JR et al. Genetic Susceptibility to Dental Caries Differs between the Sexes: A Family-based Study. 2016; 49 (2): 133-140.

2. Obradovic M et al. Caries Experience in Primary and Permanent Dentition in Children Up to 15 Years of Age from Bosnia and Herzegovina—A Retrospective Study. Child. 2023; 10 (4): 754.

4. Need to explain Model in Tables 2 and 3.

Thanks. We have added footnotes to explain the Models in Tables 2, 3, 5 and 6.

---

## [Decision Letter · Decision Letter 1]

19 Aug 2025

Early Childhood Caries Intervention in Aboriginal Australian Children: follow-up at child age 9 years

PONE-D-24-53777R1

Dear Dr. Ju,

We’re pleased to inform you that your manuscript has been judged scientifically suitable for publication and will be formally accepted for publication once it meets all outstanding technical requirements.

Kind regards,

Yolanda Malele-Kolisa, BDS, MPH, MDent, PhD

Academic Editor

PLOS ONE

Additional Editor Comments (optional):

Thank you for addressing all the comments on the manuscript.

Reviewers' comments:

Reviewer's Responses to Questions

**Comments to the Author**

1. If the authors have adequately addressed your comments raised in a previous round of review and you feel that this manuscript is now acceptable for publication, you may indicate that here to bypass the “Comments to the Author” section, enter your conflict of interest statement in the “Confidential to Editor” section, and submit your "Accept" recommendation.

Reviewer #1: All comments have been addressed

Reviewer #2: All comments have been addressed

Reviewer #3: All comments have been addressed

2. Is the manuscript technically sound, and do the data support the conclusions?

Reviewer #1: Partly

Reviewer #2: Yes

Reviewer #3: (No Response)

3. Has the statistical analysis been performed appropriately and rigorously? 

Reviewer #1: Yes

Reviewer #2: Yes

Reviewer #3: (No Response)

4. Have the authors made all data underlying the findings in their manuscript fully available?

Reviewer #1: Yes

Reviewer #2: Yes

Reviewer #3: (No Response)

5. Is the manuscript presented in an intelligible fashion and written in standard English?

Reviewer #1: Yes

Reviewer #2: Yes

Reviewer #3: (No Response)

6. Review Comments to the Author

Reviewer #1: Both groups received an intervention, so please consider changing the first sentence of the Conclusions to e.g.: "The present study suggests that, within this cohort, initiating an early childhood caries intervention during pregnancy and infancy may be associated with lower caries experience in the primary dentition by age 9 years compared to a later start."

Please also describe limitations, such as the use of a “cumulative” dmft that retains the status of exfoliated teeth previously affected by caries; while this preserves caries history, such an index can complicate interpretation at older ages.

Reviewer #2: The revised version was well improved in line with the reviewer's suggestion.

I think this manuscript should be accepted for publication in Plos One.

Reviewer #3: The study aimed to estimate the efficacy of an Early Childhood Caries (ECC) intervention among Aboriginal Australian children over 9 years, and to explore potential risk factors associated with dental caries among Indigenous Australian children.

It appeared that previous concerns were not of a statistical nature, but were requiring some clarifications which have been addressed. The design and analysis was fairly simple and well presented. Four hundred and forty-eight women pregnant with an Aboriginal child were randomly allocated to either an immediate (II) or delayed (DI) intervention group. There was no formal statistical design calculations for sample size or power in this paper. However, the sample size appeared quite adequate.

Multivariable log-Poisson regression models with robust standard error estimation were applied as a statistical model to estimate multivariable relationships of dental caries and other covariates. Risk ratios (RRs) with 95% CI were calculated. Sensitivity analyses were conducted by using the inverse-probability-of censoring weighting (IPCW) to overcome the loss-follow-up issues. Model building logically involved a hierarchical structure.

The models were appropriately applied and interpreted. The conclusions followed from the analyses performed and were appropriately presented.

7. PLOS authors have the option to publish the peer review history of their article (what does this mean?). If published, this will include your full peer review and any attached files.

Reviewer #1: No

Reviewer #2: No

Reviewer #3: No

---

## [Editor Report · Acceptance letter]

PONE-D-24-53777R1

PLOS ONE

Dear Dr. Ju,

I'm pleased to inform you that your manuscript has been deemed suitable for publication in PLOS ONE. Congratulations! Your manuscript is now being handed over to our production team.

Kind regards,

on behalf of

Prof Yolanda Malele-Kolisa

Academic Editor

PLOS ONE